# Dihydroquercetin Supplementation Improved Hepatic Lipid Dysmetabolism Mediated by Gut Microbiota in High-Fat Diet (HFD)-Fed Mice

**DOI:** 10.3390/nu14245214

**Published:** 2022-12-07

**Authors:** Mengyu Wang, Hui Han, Fan Wan, Ruqing Zhong, Yoon Jung Do, Sang-Ik Oh, Xuemeng Lu, Lei Liu, Bao Yi, Hongfu Zhang

**Affiliations:** 1State Key Laboratory of Animal Nutrition, Institute of Animal Science, Chinese Academy of Agricultural Sciences, Beijing 100193, China; 2Key Laboratory of Grassland Livestock Industry Innovation, Ministry of Agriculture and Rural Affairs, College of Pastoral Agriculture Science and Technology, Lanzhou University, Lanzhou 730020, China; 3Division of Animal Disease & Health, National Institute of Animal Science, Rural Development Administration, Wanju 55365, Republic of Korea

**Keywords:** dihydroquercetin, lipid metabolism, liver, gut microbiota, high-fat diet

## Abstract

Dihydroquercetin (DHQ) is a natural flavonoid with multiple bioactivities, including hepatoprotective effects. This study aimed to investigate whether DHQ improved lipid dysmetabolism in the body, especially in the liver, and whether there is a relationship between hepatic metabolism and altered gut flora in high-fat diet (HFD)-induced mice. HFD-induced mice were given 50 mg/kg body weight DHQ intragastrically for 10 weeks. The data showed that DHQ reduced body weight, the weight of the liver and white adipose tissue as well as serum leptin, LPS, triglyceride and cholesterol levels. RNA-seq results indicated that DHQ down-regulated lipogenesis-related genes and up-regulated fatty acid oxidation-related genes, including *MOGAT1* and *CPT1A*. Furthermore, DHQ had a tendency to decrease hepatic cholesterol contents by reducing the mRNA levels of cholesterol synthesis genes such as *FDPS* and *HMGCS1*. 16S rRNA sequencing analysis indicated that DHQ significantly decreased the richness of *Lactococcus*, *Lachnoclostridium*, and *Eubacterium_xylanophilum_group*. Correlation analysis further demonstrated that these bacteria, *Lactococcus* and *Eubacterium_xylanophilum_group* in particular, had significantly positive correlation with lipid and cholesterol synthesis genes, and negative correlation with fatty acid oxidation genes. In conclusion, DHQ could improve hepatic lipid dysmetabolism potentially by improved gut microbial community, which may be used as an intervention strategy in hepatic metabolism diseases.

## 1. Introduction

Hepatic metabolic disorder is considered as a risk factor for metabolism problems, including hyperglycemia, hyperlipidemia, obesity, and fatty liver diseases, which is increasingly common worldwide [1]. If not adequately controlled, it will further lead to the occurrence of some complications such as hypertension, atherosclerosis, and associated cardiovascular diseases, becoming a major threat to human health [2]. It was reported that silymarin was useful as a complementary alternative medicine for non-alcoholic fatty liver disease (NAFLD) and liver cirrhosis for many years worldwide [3,4]. NAFLD and liver cirrhosis were usually caused by the disorder of hepatic metabolism. Thus, we speculated that dihydroquercetin (DHQ), as the bioactive ingredient of silymarin, could potentially improve hepatic metabolic disorder.

DHQ (3,5,7,3,4-pentahydroxy flavanone), a kind of secondary metabolite, was generally distributed in Douglas fir, silybum marianum, and onions [5,6,7]. It has been used as a food supplement for milk, cheese, and beef tallow due to multiple positive effects, including anti-oxidative, anti-inflammatory, and liver-protective functions for humans and animals [8]. A vitro study in 2018 suggested that DHQ resulted in the remission of alcoholic liver steatosis with potent inhibitory function on lipogensis in HepG2 cells [9]. Finger millet extracts (FME) was reported to inhibit high-fat diet (HFD)-caused hepatic metabolic alterations and prevent liver steatosis in models of mice, and DHQ was speculated to be its predominant active ingredient in FME [10]. Moreover, a recent study has revealed that DHQ treatment could decrease the total triglyceride (TG) and cholesterol (TC) contents in hepatic tissues of HFD-induced mice [11]. However, as the liver is the major metabolic organ, how DHQ supplementation regulates hepatic metabolism-related genes was unclear. In addition, DHQ had ability to regulate gut microbiota. Recently, a preclinical trial based on in vivo study revealed that the supplementation of DHQ altered the composition of intestinal microbiota with increasing *Firmicutes* levels and decreased *Bacteroidetes* levels in dextran sodium sulfate (DSS)-caused colitis mice [12]. Our previous study also demonstrated that DHQ treatment increased the fecal *Firmicutes*/*Bacteroidetes* ratio and changed the richness of some genus such as *Lactobacillus*, *Dubosiella*, and *Bacteroidetes* in DSS-induced mice [13]. Besides, a mice study showed that DHQ could correct HFD-caused gut microbiota imbalance and increase bacteria diversity [11]. In light of this, although the previous studies have proven that DHQ could improve gut microbiota composition, the underlying mechanisms of DHQ on the association of hepatic lipid metabolism with the regulation of intestinal flora, were still unclear.

Thus, in this study, we aim to investigate the regulatory effects of DHQ on hepatic lipid metabolic genes through transcriptomics technology in HFD-induced mice. Moreover, using 16S rRNA gene sequencing method, this study sought to investigate how DHQ supplementation affected gut microbiota composition, and further revealed the relationship between lipid metabolic-related indexes and gut microbiota through correlation analysis. This study would offer theoretical evidence to develop strategic approach to prevent and treat hepatic metabolism-associated disorders, as well as provide new insights for the application of DHQ as a green additive in animal husbandry.

## 2. Materials and Methods

### 2.1. Chemicals and Diets

DHQ was acquired from Shanghai Yuanye Bio-Technology Co., Ltd. (Shanghai, China) with the purity of 99.50%. The standard chow diet (SD, research diet D12450B, 3.85 kcal/g) and the HFD (research diet D12492, 5.24 kcal/g) were provided from Beijing Keao Xieli Feed Co., Ltd. (Beijing, China). Energy intake was calculated according to the following formula:CON group: Energy intake (kcal) = Feed intake (g) × 3.85 (kcal/g)
HFD and HFD+DHQ group: Energy intake (kcal) = Feed intake (g) × 5.24 (kcal/g)

### 2.2. HPLC Analysis

Through high performance liquid chromatograph (HPLC) detection, the purity of DHQ was defined using a HPLC system (Dalian YiLiTe Technologies, Dalian, China). Elution solvents were solvent A (0.1% H_3_PO_4_ in H_2_O) and solvent B (acetonitrile). The separation was preformed on a Hyper 0DS2 C18 column (4.6 mm × 250 mm, 5 μm). Moreover, column temperature was 25 °C and detection was conducted at 286 nm. After detection, the purity of DHQ was 99.50% and showed in Appendix A.

### 2.3. Animals and Experimental Design

Male C57BL/6J (6 weeks old) mice were acquired from Peking University Health Science Center (Beijing, China). They were raised in animal house with free access to diet and water under a controlled environment at 21 ± 2 °C and a 12/12 h light/dark cycle. After a week of adaptation, mice were randomly allocated to 3 groups (n = 8/group). Mice in HFD+DHQ group were fed a HFD with gavage of 50 mg/kg body weight DHQ (dissolved in corn oil). Mice in CON and HFD groups were received with SD and HFD separately, with the same volume of corn oil intragastrically. The body weight and feed intake of them were measured every week. After 10 weeks, blood materials were obtained via orbital blooding and then the mice were killed through cervical dislocation. Liver weight and white adipose tissues weight (inguinal, epididymal, and perirenal adipose tissues) was weighed and recorded. Liver tissue was quickly taken out and frozen in liquid nitrogen for future analysis. For histopathology analysis, part of liver was cut and fixed in 4% paraformaldehyde. Feces were obtained and quickly stored in liquid nitrogen for 16S rRNA sequencing.

### 2.4. Hepatic Hematoxylin and Eosin (H&E) Staining

Part of the liver was fixed with 4% paraformaldehyde, then dehydrated and embedded in paraffin. A section was cut and placed on a slide. After deparaffinization and hydration, the sections were stained with H&E for microscopy. Images were obtained with inverted fluorescence microscope (Leica DMI3000B) (Leica, Weztlar, Germany).

### 2.5. Serum and Hepatic Parameters Analysis

After the centrifugation of blood at 3000 rpm and 4 °C for 10 min, the serum samples were collected for future biochemical analysis. The contents of total triglyceride (TG) and total cholesterol (TC) in serum and hepatic tissue, as well as serum low-density lipoprotein cholesterol (LDL-C) and high density lipoprotein cholesterol (HDL-C) were identified through automatic biochemical analyzer (fully automatic bioanalyzer BK-280, Biobase Biodustry Co., Ltd., Jinan, China). The levels of leptin, insulin, and LPS in serum were identified by ELISA assay according to the manufacturers’ manuals (Nanjing Bioengineering Institute, Nanjing, China). Moreover, the homeostasis model assessment of insulin resistance (HOMA-IR) was employed to estimate insulin resistance based on the following formula:HOMA-IR = Fasting blood glucose (mmol/L) × Fasting insulin (mIU/ L)/22.5

### 2.6. Hepatic Transcriptome Sequencing and Bioinformatics Analysis

Liver tissues from the CON, HFD, and HFD+DHQ groups were stored in liquid nitrogen and entrusted to Majorbio Bio-Pharm Technology Co., Ltd. (Shanghai, China) for transcriptomics analysis. Following the manufacturer’s instructions, total RNA was extracted with Trizol^®^ reagent (Invitrogen, Waltham, MA, USA) and genomic DNA was removed with DNase I (TaKara, Otsu, Japan). The RNA quality and purity were determined by 2100 Bioanalyser (Agilent, alo Alto, CA, USA) and Nanodrop2000 (NanoDrop Technologies, Wilmington, DE, USA). Then, sequencing library was constructed for further analysis with high-quality RNA samples (OD260/280 ≥ 1.8, OD260/230 ≥ 1.0, RIN ≥ 8.0, 28/23S > 18/16S, ≥1.6 μg). Messenger RNA is separated from total RNA using oligo (dT) beads and then segmented using fragmentation buffer. Double-stranded cDNA was synthesized by reverse transcriptase and subjected to terminal-repair, phosphorylation, and “A” base addition. After PCR for 15 cycles and quantified by TBS380, sequencing was performed with the Illumina NovaSeq 6000 sequencer (2 × 150 bp read length). CASAVA base calling was employed to transform sequencing images signal to sequencing data pattern. By SeqPrep (https://github.com/jstjohn/SeqPrep (accessed on 25 December 2021)), and Sickle (https://github.com/najoshi/sickle (accessed on 25 December 2021)) programs, the clean data reads were remained following quality assessment and low quality filtering of raw reads. Then, mapped data was acquired after aligning the clean data to the mouse genome with HISAT2 (http://ccb.jhu.edu/software/hisat2/index.shtml (accessed on 25 December 2021)) analysis. Following transcripts assembly using StringTie (http://ccb.jhu.edu/software/stringtie/ (accessed on 25 December 2021)), RSEM (http://deweylab.biostat.wisc.edu/rsem/, version 1.3.3 (accessed on 25 December 2021)) software was used for quantification based on transcripts per million reads (TPM). Differential expression genes (DEGs) between groups were determined with DESeq2. Genes were defined to be significantly differentially expressed with *p*-adjust < 0.05 and fold change (FC) ≥ 2 (for up-regulation) or ≤0.5 (for down-regulation).

### 2.7. Quantitative Real-Time PCR (qRT-PCR) Analysis

Biochemical reagents, including Trizol (Invitrogen, USA) reagent, chloroform, isopropanol, and 75% ethanol solution, were used to isolate total RNA of liver tissues. Then, RNA concentrations were detected with the NanoDrop 2000 (Nanodrop Technologies, USA). After removing the possible DNA contaminations, the reverse transcription was conducted using PrimeScript RT reagent kits (Takara, Beijing, China). Next, qRT-PCR was conducted in accordance with the manual of the manufacturer with KAPA SYBR FAST qPCR Master Mix kit. Briefly, reactions were carried out in triplicate for each sample in a 10 μL volume consisting of 5 μL KAPA SYBR FAST qPCR Master Mix Universal, 0.4 μL PCR forward primer, 0.4 μL PCR reverse primer, 0.2 μL ROX low, 1 μL cDNA, and 3 μL PCR-grade water following the instruction of the manufacturer (KAPA biosystems, Beijing, China). β-actin was designed as internal control and each result of target genes was normalized to β-actin and relatively quantified with 2^−ΔΔCt^ method. The RT-PCR was performed on an Applied Biosystems 7500 RT-PCR System (Thermo Fisher Scientific, Shanghai, China). Specific primers were synthesized in Sangon Biotech Co., Ltd. (Shanghai, China) and exhibited in Table 1.

### 2.8. Gut Microbiota Profiling in Feces

Microbial community total genome DNA of feces was extracted with the FastDNA^®^ Spin Kit for Soil (MP Biomedicals, Solon, OH, USA). Then, the quality of total genome DNA was analyzed with 1% agarose gel. Its purity and concentration were identified by NanoDrop 2000 UV-vis spectrophotometer (Thermo Scientific, Wilmington, DE, USA). PCR amplification of hypervariable region V3-V4 in the bacteria 16S ribosomal RNA gene was performed using specific primers (338F, 5′-ACTCCTACGGGAGGCAGCAG-3′; 806R, 5′-GGACTACHVGGGTWTCTAAT-3′) with an ABI GeneAmp^®^ 9700 PCR thermocycler (ABI, Los Angeles, CA, USA). The PCR product was checked with 2% agarose gel, followed with purification, quantification, and homogenization with AxyPrep DNA Gel Extraction Kit (Axygen Biosciences, Union City, CA, USA) and Quantus™ Fluorometer (Promega, Fitchburg, WI, USA) based on the manufacturer’s requirements. After that, purified amplicons were sequenced with an Illumina MiSeq PE300 platform (Illumina, San Diego, CA, USA) at Majorbio Bio-Pharm Technology Co., Ltd. (Shanghai, China) based on the standard protocols. The sequences were quality-filtered and merged, then clustered to operational taxonomic units (OTUs) with 97% similarity using the UPARSE software (http://www.drive5.com/uparse/, version 7.1 (accessed on 25 December 2021)). Alpha diversity, including Simpson, Shannon, and Chao indices, was conducted by the Mothur version 1.30.2. Beta diversity was determined through calculating the weighted Unifrac distance and visualized by principal co-ordinates analysis (PCoA).

### 2.9. Correlation Analysis

Pearson’s correlation analysis was carried out between gut microbiota and hepatic lipid metabolic parameters.

### 2.10. Statistical Analysis

Statistical analyses were processed through one-way analysis of variance (ANOVA) with Duncan test (SPSS 22.0 software, IBM Corp., Chicago, IL, USA). The data were expressed as the mean ± SEM. *p* < 0.05 was considered to be significant statistically.

## 3. Results

### 3.1. DHQ Alleviated the Weight Parameters and Prevented Fat Accumulation in Hepatic Tissue in HFD-Induced Mice

To establish a model of lipid metabolic disorder, the mice were raised with HFD for consecutive 10 weeks (Figure 1A). The results showed that there was higher final body weight in the HFD group compared with the CON group, whereas DHQ supplementation to HFD-fed mice prevented weight gain, leading to a significantly lower body weight after 10 weeks (*p* < 0.05) (Figure 1B,C). During the whole experiment, HFD-fed mice showed the decreased average daily food intake and increased energy inputs than mice fed with a SD. Meanwhile, the average daily food consumption and energy inputs was similar between HFD and HFD+DHQ groups (Figure 1D,E). Furthermore, mice fed with a HFD displayed a significant elevation of the liver weight than CON mice, which could be alleviated by the administration of DHQ (*p* < 0.05) (Figure 1F). Compared to the normal controls, HFD-fed mice exhibited significant higher weight of inguinal, epididymal, and perirenal adipose tissue, which was dramatically reversible by administration of DHQ (*p* < 0.05). Meanwhile, HFD increased the ratios of the weight of inguinal, epididymal, and perirenal adipose tissue to the body weight (*p* < 0.05). However, supplementation with DHQ significantly decreased inguinal and perirenal fat indices, while failed to affect epididymal fat indices (Figure 1F,H).

### 3.2. DHQ Alleviated Lipid Metabolism-Related Biomarkers in Serum of HFD-Induced Mice

Mice fed with a HFD presented a significant increase of leptin, and insulin concentrations, accompanied by a notable increase in HOMA-IR than the control, which were the hallmark features of leptin, and insulin resistance as well as metabolic disorders. However, the supplementation of DHQ could reduce leptin level significantly (*p* < 0.05), and tend to reduce insulin level and HOMA-IR, as shown in Figure 2A–C (*p* > 0.05). Moreover, serum LPS levels were significantly increased in mice fed with a HFD than normal group, which could be reversed by DHQ (*p* < 0.05) (Figure 2D). Simultaneously, in comparison with CON group, HFD-induced mice had elevated TG and TC contents in serum, which could be alleviated by DHQ treatment (*p* < 0.05) (Figure 2E,F). Besides, the anomalous increase in serum HDL-C caused by HFD was significantly inhibited by DHQ supplementation (*p* < 0.05). There was no significant difference in LDL-C contents of serum among CON, HFD, and HFD+DHQ groups (Figure 2G,H).

### 3.3. DHQ Improved Lipid Accumulation in Livers of HFD-Induced Mice

As a critical organ for body metabolism, the liver is susceptible to high fat or high energy diet. In this study, pathological observations showed that HFD resulted in the hepatic abnormal accumulation of lipid droplets, which was suppressed after DHQ treatment (Figure 3A). Consistent with this, hepatic TG and TC levels were significantly higher in the mice of the HFD group compared with the CON group (Figure 3B,C). In contrast, the supplementation of DHQ significantly decreased the TG level (*p* < 0.05), and had a tendency to decrease the TC level (*p* > 0.05).

### 3.4. Identification of Differentially Expressed Genes (DEGs) in Liver

Considering that DHQ intervention could attenuate hepatic lipid deposition caused by HFD, we next investigated the effect of DHQ on the mRNA levels of hepatic genes associated with lipid metabolism based on RNA sequencing methods. As shown in Table 2, about 57.6, 53.3, and 55.4 million raw reads were acquired from the CON, HFD, and HFD+DHQ groups, respectively. Through filtering and removing contamination of the raw reads, high-quality clean reads (57.0, 52.8, and 54.9 million in the CON, HFD, and HFD+DHQ groups, respectively) were obtained. Moreover, the average quality of Q20 and Q30 bases in all 12 samples surpassed 98% and 94%, suggesting excellent sequencing quality. After mapped to mouse reference genome of the clean reads, the mapping ratios in three groups all exceed 86%, meeting the requirement for assembly and further analysis.

The principal component analysis (PCA) result demonstrated that the samples of the HFD group were significantly separated from the CON group, while the gene clustering from the HFD+DHQ group was closer to the CON group (Figure 4A). A strict comparison at *p*-adjust < 0.05 and FC ≥ 2 (for up-regulation) or ≤−2 (for down-regulation) was conducted to define the number of DEGs. Volcano plots were generated to visualize the distribution of DEGs in the CON and HFD, the HFD and HFD+DHQ groups. There were 577 DEGs (240 being up-regulated and 337 being down-regulated) were determined in the HFD-fed mice than the CON mice. Meanwhile, 31 genes in total have been significantly differentially expressed with 20 being up-regulated and 11 being down-regulated in the HFD+DHQ group, compared to the HFD group (Figure 4B,D).

### 3.5. DEGs Expression Patterns and Functional Annotation from RNA-Seq in the Liver

To characterize the functional results of DEGs provoked by the HFD and DHQ, KEGG and GO analysis was performed. By matching altered genes to the KEGG database, DEGs were functionally annotated in metabolism, human diseases, and organismal systems. As shown in Figure 5A, DEGs in the CON and HFD groups were the most enriched in metabolism, including retinol metabolism, steroid hormone biosynthesis, and drug metabolism-cytochrome P450. Moreover, chemical carcinogenesis was the most enriched in human diseases term, as the inflammatory mediator regulation of TRP channels was the most enriched in organismal systems term. In contrast, for altered gene sets in the HFD and HFD+DHQ groups, metabolism, human diseases, organismal systems, and environmental information processing were the most abundant functional groups. Metabolism and environmental information processing terms, including starch and sucrose metabolism, arachidonic acid metabolism, and the FoxO signaling pathway, were significantly enriched, as human diseases and organismal systems terms such as the adipocytokine signaling pathway, pancreatic secretion, central carbon metabolism in cancer, and pathways in cancer were enriched in our results (Figure 5B). A further investigation based on GO enrichment analysis showed several enriched processes such as the cholesterol biosynthetic process, the regulation of the triglyceride metabolic process, and the long-chain fatty acid metabolic process involved in lipid metabolism in DEGs from the CON and HFD group (Figure 5C). However, DHQ mainly altered the genes associated with glucan metabolic process, glycogen metabolic process, and glucose homeostasis (Figure 5D).

### 3.6. DHQ Altered the Hepatic Expression Levels of Genes Involved in Triglyceride and Cholesterol Metabolism in HFD-Induced Mice

Based on the above-mentioned data, we next focused on the genes associated with lipid metabolic processes and identified hub genes that had potential influence on the results. Figure 5A showed that HFD up-regulated the mRNA levels of *1-acylglycerol-3-phosphate O-acyltransferase 1* (*AGPAT1*), *diacylglycerol O-acyltransferase 2* (*DGAT2*), *monoacylglycerol O-acyltransferase 1* (*MOGAT1*), *Mid1 interacting protein 1* (*MID1IP1*), *acyl-CoA thioesterase 11* (*ACOT11*), *fatty acid synthase* (*FASN*), *sterol regulatory element binding transcription factor 1* (*SREBF1*), and *MLX interacting protein*-*like* (*MLXIPL*) involved in lipogenesis, which was reversed by DHQ supplementation. Meanwhile, the expression of *carnitine palmitoyltransferase 1a* (*CPT1A*), *enoyl*-*CoA*, *hydratase*/3-*hydroxyacyl CoA dehydrogenase* (*EHHADH*), *long*-*chain acyl*-*Coenzyme A dehydrogenase* (*ACADL*), *CYP4A10*, *CYP4A14*, and *CYP4A31* related to fatty acid oxidation were decreased in mice fed with a HFD than those of normal diet, which also could be reversed with DHQ. Moreover, HFD and DHQ altered the mRNA expression of genes associated with lipid transport, including *CD36 molecule* (*CD36*), *solute carrier family 27*, *member 5* (*SLC27A5*), and *microsomal triglyceride transfer protein* (*MTTP*) (Figure 6A).

In addition, DHQ reprogramed some gene sets involved in cholesterol metabolism. Specifically, HFD increased the mRNA levels of *acetyl-CoA acetyltransferase 2* (*ACAT2*), *farnesyl diphosphate synthetase* (*FDPS*), *3-hydroxy-3-methylglutaryl-CoA synthase 1* (*HMGCS1*), *3-hydroxy-3-methylglutaryl-CoA reductase* (*HMGCR*), *isopentenyl-diphosphate delta isomerase* (*IDI1*), *lanosterol synthase* (*LSS*), *methylsterol monooxygenase 1* (*MSMO1*), *mevalonate diphosphate decarboxylase* (*MVD*), *NAD(P) dependent steroid dehydrogenase-like* (*NSDHL*), *squalene epoxidase* (*SQLE*), *acetoacetyl*-*CoA synthetase* (*AACS*), and *sterol-C5-desaturase* (*SC5D*) involved in cholesterol synthesis, while *ACAT2*, *FDPS*, *HMGCS1*, *MVD*, *NSDHL*, *AACS*, and *SC5D* levels were down-regulated after DHQ administration to HFD-fed mice. Notably, DHQ also changed the genes related to cholesterol transport and cholesterol homeostasis such as *peroxisome proliferator activated receptor gamma* (*PPARG*), *ATP binding cassette subfamily G member 1* (*ABCG1*), *apolipoprotein A-IV* (*APOA4*), *CYP39A1*, *neutral cholesterol ester hydrolase 1* (*NCEH1*), and *sterol O-acyltransferase 2* (*SOAT2*) (Figure 6B). Besides, Figure 5C showed mevalonate pathway in cholesterol synthesis.

### 3.7. Verification of RNA-Seq Results through qRT-PCR in the Liver

To prove the correctness of the results obtained from RNA-seq analysis, qRT-PCR was employed to analyze the mRNA expression of nine genes selected randomly, including *phosphoenolpyruvate carboxykinase 1* (*PCK1*), *lipin 2* (*LPIN2*), *glycerophosphocholine phosphodiesterase 1* (*GPCPD1*), *CYP4A32*, *abhydrolase domain containing 2* (*ABHD2*), *sphingomyelin phosphodiesterase 3* (*SMPD3*)*, elongation of very long chain fatty acid elongase 2* (*ELOVL2*), *Fibrinogen-like protein 1* (*FGL1*), and *CPT1A*. Relative to the CON group, several genes were markedly down-regulated in the HFD group, including *PCK1*, *LPIN2*, *GPCPD1*, *CYP4A32*, *ABHD2*, *SMPD3*, *ELOVL2*, *FGL1*, and *CPT1A* (*p* < 0.05). Meanwhile, it was found that DHQ significantly enhanced the expression of *LPIN2*, *GPCPD1*, *CYP4A32*, *ELOVL2*, and *FGL1* for mice induced by HFD (*p* < 0.05) (Figure 7A). In brief, the expression trends of the detected genes were highly in conformance with the results obtained by RNA-seq (Figure 7B,C). Differences in fold change values were likely due to the different detection sensitivity of the two methods [14].

### 3.8. DHQ Modified Gut Microbiota Composition in HFD-Induced Mice

Next, 16S rRNA gene sequencing was used to identify microbiota structure in the feces with 7–8 biological replicates in each group. As shown in Figure 8A, the rarefaction curves indicated that the end of the curve tends to be flat, demonstrating that the amount of sequencing data is reasonable enough to reflect the vast majority of microbiota diversity in fecal samples. The Venn diagram presented 336 common OUTs and 36, 43, and 42 individual OTUs, respectively, in the CON, HFD, and HFD+DHQ groups based on 97% sequence similarity (Figure 8B). Further, the Simpson, Shannon, and Chao indexes were determined for alpha-diversity and community richness of gut flora. As illustrated in Figure 8C, daily HFD treatment for 10 weeks significantly reduced the Simpson index (*p* < 0.05), while displayed tenuous effect on the Shannon and Chao indexes than CON group. Moreover, DHQ had no significant impact on Simpson, Shannon, and Chao indexes in the HFD-fed mice. Then, β-diversity was evaluated via principal co-ordinates analysis (PCoA) based on Bray-Curtis distance. It was showed that the CON group showed an obvious separation in microbiota structure compared with the HFD and HFD+DHQ groups. Meanwhile, there is also a discrepancy in the make-up of gut bacteria between the HFD and HFD+DHQ groups, revealing that the gut micorbial community in mice fed with a HFD was influenced by DHQ (Figure 8D). 

At the phylum level, compared with the CON group, the supplementation of the HFD decreased the richness of *Firmicutes* significantly (*p* < 0.05), which can be reversed by DHQ, although not significantly (*p* > 0.05) (Figure 8E). In addition, there was no significant alterations in the relative abundance of *Bacteroidetes* among the CON, HFD, and HFD+DHQ groups (Figure 8E). At the genus level, the proportions of *Streptococcus*, *Lactococcus*, *Enterococcus*, *unclassified_f_Lachnospiraceae*, *Lachnoclostridium*, and *Eubacterium_xylanophilum_group* was significantly increased, whereas *Coriobacteriaceae_UCG-002* was significantly decreased in mice of the HFD group than those of normal controls (*p* < 0.05). Noteworthy, the alterations were reversed after DHQ supplementation. In particular, the richness of *Lactococcus*, *Lachnoclostridium*, and *Eubacterium_xylanophilum_group* was significantly reduced by DHQ in HFD-caused mice (Figure 9).

### 3.9. Correlation Analysis between the Altered Gut Microbiota and Lipid Metabolic Indexes

The relevance of the altered gut microbiota and lipid metabolism-associated indicators were evaluated by Pearson’s correlation analysis and visualized through heatmap (Figure 10 and Appendix A). As demonstrated in Figure 10, the relative abundance of *Streptococcus* was significantly positively associated with serum insulin, LPS, TC, and hepatic TG levels. The richness of *Lactococcus* and *Enterorcoccus* had positive relationship with body weight, serum leptin, insulin, LPS, TC, and hepatic TG levels. Moreover, the relative richness of *unclassified_f_Lachnospiraceae* and *Eubacterium_xylanophilum_group* showed positive correlation with the amounts of leptin, TC in serum, and TG in liver. Furthermore, the relative abundance of *Lachnoclostridium* correlated positively with body weight, serum leptin, LPS, TG, and hepatic TG, TC levels. Besides, it is clear that the relative abundance of *Coriobacteriaceae_UCG-002* had negative correlation with serum LPS, TG and hepatic TG levels. Meanwhile, the correlation analysis between flora and key genes in the hepatic transcriptome data was performed. The results indicated that the relative abundance of *Streptococcus* and *Eubacterium_xylanophilum_group* was significantly positively related to *MOGAT1*, *MID1IP1*, *ACAT2*, *FDPS*, and *MVD*, but significantly negatively correlated with *CPT1A* expression. The richness of *Lactococcus* and *Enterorcoccus* showed positive correlation with the expression of *DGAT2*, *MOGAT1*, *MID1IP1*, *MLXIPL*, *ACAT2*, *FDPS*, *HMGCS1*, *MVD*, and *NSDHL*, but negative correlation with *CPT1A* and *CYP4A10* expression. Furthermore, there was a positive relationship between the richness of *unclassified_f_Lachnospiraceae* and the expression of *FDPS*, *HMGCS1*, *MVD* and *NSDHL* in the liver. In addition, the proportion of *Lachnoclostridium* had positively correlation with *MID1IP1*, *FDPS*, *HMGCS1*, and *MVD* mRNA levels.

## 4. Discussion

Lipid metabolic disorder is emerging as an epidemic pathological process with promoting the incidence and progression of fatty liver diseases, seriously endangering people’s health [15]. In the present study, we found that DHQ supplementation improved hepatic lipid dysmetabolism with decreasing lipid and cholesterol synthesis-associated genes as well as increasing fatty acid oxidation-associated genes, which was related to the improved gut microbiota composition.

Lipid metabolism imbalance induced by HFD is often concomitant with the increase of body weight and fat storage [16,17]. A recent study has indicated that DHQ supplementation had beneficial metabolic effects for HFD-fed mice, which is featured by decreasing body weight and fat accumulation [11]. Similarly, our data showed that dietary HFD caused increased body weight and adipose tissue weight (inguinal, epididymal, and perirenal adipose tissue), which was remarkably decreased by DHQ treatment. Moreover, HFD feeding usually caused hyperlipidemia, which were considered as risk indicators for metabolic disorder [18]. Here, the consumption of a HFD resulted in higher levels of serum TG and TC, which was coincident with an earlier study [19]. It was reported that elevated TG and TC levels in serum could promote the development of NAFLD and other metabolic diseases [20]. Notably, the present study showed that DHQ supplementation had significantly reduced serum TG and TC production in mice fed with a HFD. HDL-C is involved in transporting cholesterol to the liver from peripheral tissues and decomposing it into bile acids, which in turn are released into the intestine and excreted from the body [21]. In this work, we also found that DHQ supplementation inhibited the abnormal increase of serum HDL-C level caused by a HFD. Collectively, these data demonstrated that DHQ supplementation decreased body fat storage and ameliorated hyperlipidemia in mice fed with a HFD.

Leptin, an adipocyte-secreted hormone, is required for maintaining energy balance through mediating processes involved in food intake and expenditure [22]. Insulin displayed important function in regulating lipid metabolism. During the challenge of insulin resistance, insulin promoted de novo lipogenesis and re-esterification in the liver [23]. HFD-fed models usually had enhanced production of systemic leptin and insulin, which was associated with leptin and insulin resistance, as well as metabolic disorder [24,25,26]. A partial reduction of leptin content can promote weight loss and mitigate both leptin and insulin resistance in HFD-fed mice [27]. In the current study, it was revealed that DHQ supplementation decreased serum leptin level and had a tendency to reduce serum insulin level and HOMA-IR, which further revealed the beneficial impacts of DHQ on body metabolism in HFD-induced mice.

When fed with long-term high fat or carbohydrate diets, lipids are generally deposited in hepatic tissue, resulting in metabolic liver dysfunction, and eventually developing into NAFLD or hepatic fibrosis [28]. In the present study, the liver weight was measured, and the HFD significantly increased the liver weight, which was in accordance with a previous study [20]. The histological results from H&E staining also confirmed that massive lipid droplets were distributed in the livers of HFD-fed mice here. Furthermore, according to our data, the HFD-fed mice showed increased hepatic TG and TC contents. These data demonstrated that a HFD caused excess lipid deposition in the liver tissue in this work. Nevertheless, after DHQ supplementation, the liver weight, hepatic TG content and lipid droplets numbers were significantly reduced, implying that DHQ treatment played a positive role in alleviating abnormal lipid accumulation in hepatic tissue. Taken together, these results indicated that DHQ supplementation effectively improved lipid deposition in the livers of HFD-fed mice.

As a principal regulator of lipid metabolism, the liver is implicated in modulating lipid synthesis and utilization, fatty acid metabolism, and cholesterol homeostasis, etc. [29]. Previous studies described that HFD can cause hepatic lipid accumulation by modulating lipid metabolism-related genes [30,31]. In parallel, numerous bioactive flavonoids from natural plants have been confirmed to attenuate HFD-induced lipid deposition in the liver via regulating the genes related to lipogensis and fatty acid metabolism [32,33,34]. Therefore, this experiment further investigated the mechanism of DHQ-reducing hepatic lipids in HFD-fed mice from the level of mRNA levels. The results showed that the HFD-fed mice had higher expression of *AGPAT1*, *DGAT2*, *MOGAT1*, *MID1IP1*, *ACOT11*, *FASN*, *SREBF1*, and *MLXIPL* in the liver required from RNA-seq analysis. AGPAT1, DGAT2, and MOGAT1 are implicated in triglyceride synthesis, while MID1IP1, ACOT11, and FASN are associated with fatty acid synthesis [35,36,37]. The proteins encoded by *SREBF1* and *MLXIPL* played important effects in promoting the transcription of lipogenic genes, such as *FASN* [38]. However, after DHQ treatment, these genes in liver were down-regulated. In especial, *MOGAT1*, *MID1IP1*, and *ACOT11* mRNA levels in hepatic tissue were greatly lessened by DHQ in mice fed with a HFD (fold changes were 0.632, 0.697, and 0.476, respectively). MOGAT1 is involved in converting monoacylglycerol to diacylglycerol, which was overexpressed in the livers of obese mice suffering from hepatic steatosis [39]. It was reported that *MOGAT1* gene deficiency decreased weight gain and hepatic fat mass as well as improved glucose tolerance and insulin signaling in obese mice fed with a diet containing high energy [40]. MID1IP1 was associated with the synthesis of fatty acid, which promote the risk of fatty liver diseases [41]. ACOT11, a member of the acyl-CoA thioesterase family, plays a critical role in catalyzing the hydrolysis of fatty acyl CoA to release free fatty acids and CoA, which is regulated by ambient temperature and food expenditure. Knockdown of ACOT11 could successfully combat HFD-induced obesity and hepatic steatosis [42]. Here, the reduction of *MOGAT1*, *MID1IP1*, and *ACOT11* expression caused by DHQ supplementation may be greatly responsible for the decrease of triglyceride content in hepatic tissue. Furthermore, various genes, including *ACADL*, *CPT1A*, *EHHADH*, *CYP4A10*, *CYP4A14*, and *CYP4A31*, were down-regulated in mice fed with a HFD, whereas DHQ supplementation inhibited these changes. CPT1A, EHHADH, and ACADL are implicated in β-oxidation of fatty acid, while CYP4A10, CYP4A14, and CYP4A31 were involved in ω-oxidation of fatty acid [43,44,45]. Of note, CPT1A is the rate-limiting enzyme in fatty acid oxidation, which is responsible for transporting fatty acids from cytoplasm to mitochondria for oxidative decomposition. Here, the elevated expression of *CPT1A*, *EHHADH*, *ACADL*, *CYP4A10*, *CYP4A14*, and *CYP4A31* resulting from DHQ supplementation contributed to the enhancement of fatty acid oxidation, which could further explain the smaller hepatic TG accumulation in mice fed with a HFD. CD36 and SLC27A5 are fatty acid translocases capable of mediating fatty acid uptake into the liver [46]. In the current study, DHQ supplementation down-regulated the expression of *CD36* and *SLC27A5* in the livers of HFD-fed mice, which promoted the decrease of fatty acid uptake in hepatic tissue. Altogether, these results suggested that DHQ reduced triglyceride level and alleviated lipid deposition in hepatic tissue potentially through regulating adipogenesis, fatty acid oxidation, and fatty acid transport-related genes.

In addition to the beneficial effect on triglyceride homeostasis, cholesterol metabolism was also affected by DHQ in this study. In fact, it was shown that excess lipid accumulation in the liver could disturb cholesterol metabolism, causing the dysregulation of cholesterol synthetic pathways, bile acid metabolism, and cholesterol export [47,48]. In the current study, RNA-seq results demonstrated that HFD up-regulated the expression of *ACAT2*, *FDPS*, *HMGCS1*, *HMGCR*, *IDI1*, *LSS*, *MSMO1*, *MVD*, *MVK*, *NSDHL*, *SQLE*, *AACS*, and *SC5D*, which were involved in de novo cholesterol synthesis [49,50]. However, after DHQ treatment, the expression of *ACAT2*, *FDPS*, *HMGCS1*, *MVD*, *NSDHL*, *AACS*, and *SC5D* was down-regulated, which promoted cholesterol decrease in mice fed with a HFD here. Moreover, genes associated with cholesterol transport such as *PPARG* were altered by HFD feeding, whereas these alterations were reverted following DHQ treatment. PPARG is known to be implicated in cholesterol efflux and helps maintain intracellular cholesterol homeostasis [51]. In the present study, DHQ activated the expression of *PPARG*, which partially maybe explain the reason for the decreased cholesterol level in the livers of HFD-fed mice. CYP39A1 was expressed in the hepatic endoplasmic reticulum and participated in bile acid synthesis [52], which is the most important mechanism of cholesterol degradation in the liver [53]. Here, DHQ up-regulated the expression of *CYP39A1* in HFD-fed mice, which produced positive activities on cholesterol clearance. Altogether, these results suggested that administration of DHQ could improve cholesterol homeostasis in hepatic tissues of HFD-fed mice, mainly through mediating cholesterogenic genes. Notably, the reliability of RNA-seq data was verified by qRT-PCR for randomly selected genes.

Indeed, gut bacteria had a crucial implication for the regulation of hepatic lipid metabolism [54]. Previous studies have shown that the disruption of gut microbiota balance could easily induce hepatic lipid dysmetabolism and even resulted in fatty liver-associated diseases [55,56]. Interestingly, preclinical trials have shown that flavonoids supplementation from natural plants such as mulberry leaves and hawk tea, could ameliorate hepatic lipid metabolism in association with the changes of gut microbiota [57,58]. Thus, the fecal microbial was analyzed by 16S rRNA sequencing in the present study. We observed a significant elevation in the proportion of *Firmicutes* in feces collected from the HFD-fed mice, which was in agreement with many former studies [59,60]. However, DHQ treatment on the HFD-fed mice could inhibit this change, although not significantly. At the genus level, HFD feeding enriched the relative abundance of *Lachnoclostridium* and *unclassified_f_Lachnospiraceae* within the family *Lachnospiraceae*, which was potentially pathogenic bacteria and reported to be linked with diet-induced metabolic diseases in humans and mouse models [61,62,63]. Zhao et al. found that the decreased abundance of *Lachnospiraceae* was involved in the alleviation of alcoholic fatty liver [64]. Similarly, in this experiment, DHQ administration significantly reduced the relative abundance of *Lachnoclostridium* and tended to reduce *unclassified_f_Lachnospiraceae* richness, which contributed to the improvement of hepatic lipid metabolism. Moreover, in the HFD-fed mice, DHQ significantly suppressed the increased amount of sequence assigned to *Eubacterium_xylanophilum_group*, which was abundant in obese mice according to a previous study [65]. This means that *Eubacterium_xylanophilum_group* may be a potential bacterium related to metabolic disorder, but its role is unclear and needs further exploration. Meanwhile, HFD significantly lowered the richness of *Coriobacteriaceae_UCG_002*, which is a member of the *Coriobacteriaceae* family and had a negative effect on cholesterol and triglycerides homeostasis [66]. It has been originally suggested that *Coriobacteriaceae_UCG_002* elevation was associated with the development of NAFLD [67]. Surprisingly, DHQ supplementation had a tendency to increase the *Coriobacteriaceae_UCG_002* richness. We speculated that different experimental period and materials might result in this inconsistent result due to the susceptibility and dynamic changes of intestinal flora. Additionally, the proportion of *Streptococcus*, *Lactococcus*, and *Enterococcus* was abundant in the HFD-fed mice. When treated with DHQ, *Lactococcus* richness was significantly decreased, as *Streptococcus* and *Enterococcus* richness displayed a decreasing trend. They were identified as LPS-producing bacteria [59,68,69]. Consistently, in this work, serum LPS level was increased in HFD-fed mice. Meanwhile, after DHQ supplementation, the LPS level in serum was significantly decreased in mice fed with a HFD, which might be due to the decreased *Streptococcus*, *Lactococcus*, and *Enterococcus* relative abundance [70]. Taken together, these results suggested that the administration of DHQ can improve the composition of gut microbiota by decreasing the harmful bacteria, especially *Lachnoclostridium*, *Lactococcus* and *Eubacterium_xylanophilum_group*.

To further explain whether the DHQ-alleviated hepatic lipid metabolic dysregulation was associated with gut microbiota, Pearson’s correlation analysis was performed between lipid metabolic parameters and altered intestinal bacteria. In this work, the results showed that the relative abundance of *Streptococcus*, *Lactococcus*, and *Enterorcoccus* had significant positive correlation with serum insulin, LPS, TC, and hepatic TG contents. *Streptococcus*, *Lactococcus*, and *Enterorcoccus* were recognized as the predominant genus in subjects with obesity or fatty liver disorders [59,68,69]. Moreover, Tang et al. demonstrated that *unclassified_f_Lachnospiraceae* and *Lachnoclostridium* was relevant for the increase of hepatic TG content [71]. In keeping with this observation, the current study showed that the relative abundance of *unclassified_f_Lachnospiraceae* and *Lachnoclostridium* had a positive relationship with serum leptin and hepatic TG levels. As described in previous findings, the relative abundance of *Eubacterium_xylanophilum_group* had a positive correlation with body mass index and HOMA-IR [65], suggesting that this genera plays a beneficial role in regulating body lipid metabolism. Consistently, the *Eubacterium_xylanophilum_group* was here found to be positively correlated with the content of leptin, TC in serum, and TG in the liver. Simultaneously, the correlation analysis between flora and key genes associated with hepatic lipid metabolism from transcriptome data was conducted. It was shown that the relative abundance of *Streptococcus* had a positive correlation with lipogenic genes *MOGAT1*, *MID1IP1*, and *MLXIPL*, but had a negative correlation with fatty acid oxidation genes *CPT1A* and *CYP4A10*. Moreover, *Lactococcus* and *Enterorcoccus* were positively correlated with *DGAT2*, *MOGAT1*, *MID1IP1*, and *MLXIPL*, but negatively correlated with *CPT1A* and *CYP4A10*. Meanwhile, we also found that *Eubacterium_xylanophilum_group* richness showed a positive relationship with *MOGAT1* and *MID1IP1* expression, while it was negatively correlated with *CPT1A*. This means that *Streptococcus*, *Lactococcus*, *Enterorcoccus*, and the *Eubacterium_xylanophilum_group* are pivotal bacteria that promoted lipid accumulation in the liver. Furthermore, *Lactococcus*, *Enterorcoccus*, and *Eubacterium_xylanophilum_group* had positive correlation with cholesterol synthesis genes *ACAT2*, *FDPS*, *HMGCS1*, *MVD*, and *NSDHL* mRNA levels. Besides, the relative abundance of *unclassified_f_Lachnospiraceae* was positively correlated with *FDPS*, *HMGCS1*, *MVD*, and *NSDHL* expression and *Lachnoclostridium* was positively correlated with *FDPS*, *HMGCS1*, and *MVD* expression in the liver. The data proved that these bacteria might produce adverse effects on cholesterol homeostasis and contribute to excess hepatic cholesterol accumulation, especially *Lactococcus*, and the *Eubacterium_xylanophilum_group*. Collectively, the new information here contributed to understanding the improved hepatic lipid metabolism followed by DHQ supplementation, at least in part, was mediated by the alterations of gut microbiota.

## 5. Conclusions

In the present study, we showed that the treatment of DHQ inhibited the synthesis of lipids and cholesterol, promoted fatty acid oxidation, which was associated with the reduced deleterious bacteria, especially *Lactococcus* and the *Eubacterium_xylanophilum_group*. In conclusion, these results demonstrated that DHQ has the potential to be used as an effective dietary supplement in preventing and treating hepatic lipid metabolism disorders, providing a possible approach to human therapy for fatty liver.

## Figures and Tables

**Figure 1 nutrients-14-05214-f001:**
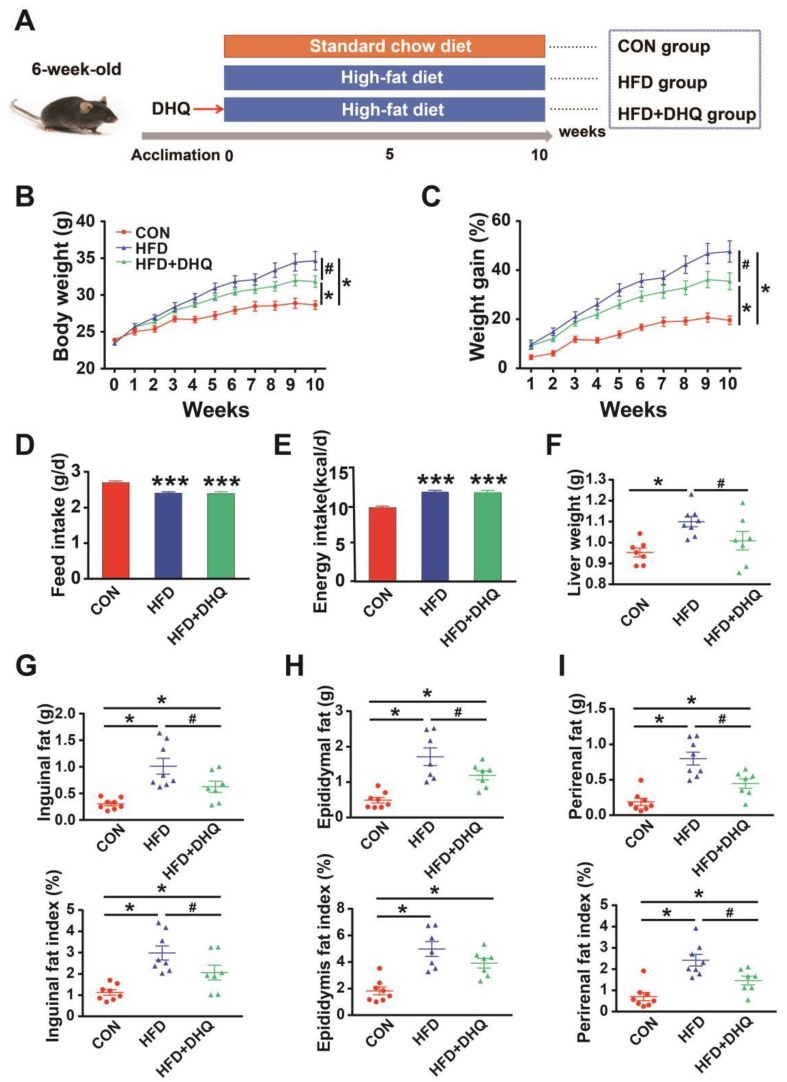
Effects of DHQ on growth performance and lipid deposition in mice. (**A**) Animal breeding scheme (three groups); (**B**) body weight; (**C**) weight gain; (**D**) feed intake; (**E**) energy intake; (**F**) liver weight; (**G**) inguinal fat weight and index; (**H**) epididymal fat weight and index; (**I**) perirenal fat weight and index. Data were expressed as the mean ± SEM. * *p* < 0.05, *** *p* < 0.001 vs. CON group, ^#^ *p* < 0.05 vs. HFD group.

**Figure 2 nutrients-14-05214-f002:**
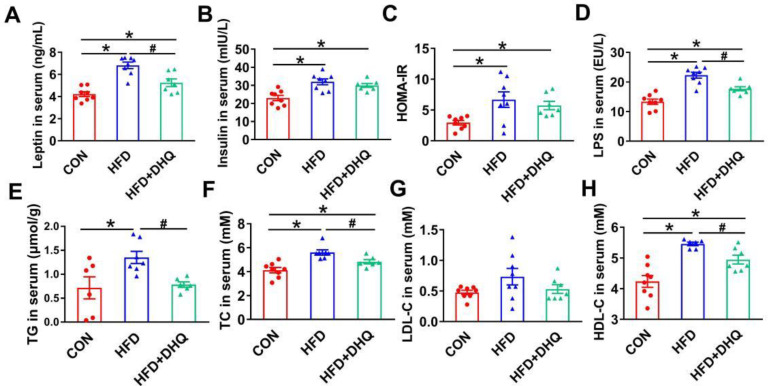
Effects of DHQ on serum parameters in mice. (**A**) Leptin; (**B**) insulin; (**C**) HOMA-IR; (**D**) LPS; (**E**) TG; (**F**) TC; (**G**) LDL-C; and (**H**) HDL-C. Data were expressed as the mean ± SEM. * *p* < 0.05 vs. CON group, ^#^ *p* < 0.05 vs. HFD group.

**Figure 3 nutrients-14-05214-f003:**
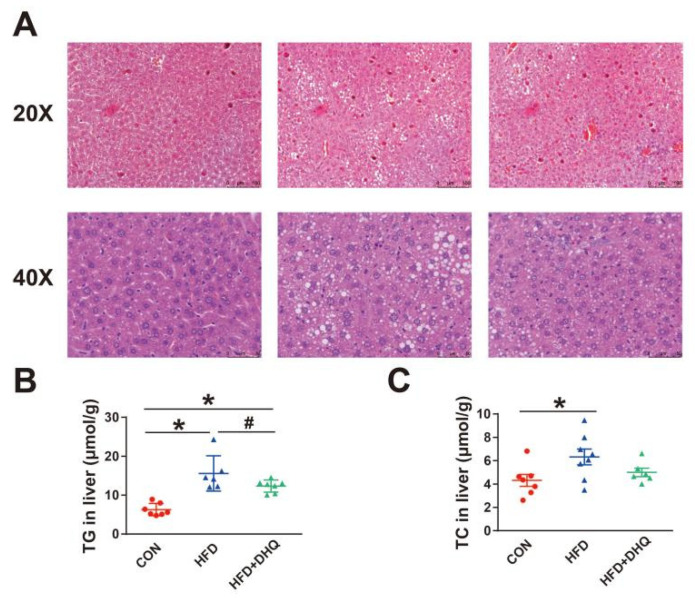
Effects of DHQ on lipid deposition in livers of mice. (**A**) H&E staining images of hepatic tissue; (**B**) TG level; and (**C**) TC level. Data were expressed as the mean ± SEM. * *p* < 0.05 vs. CON group, ^#^ *p* < 0.05 vs. HFD group.

**Figure 4 nutrients-14-05214-f004:**
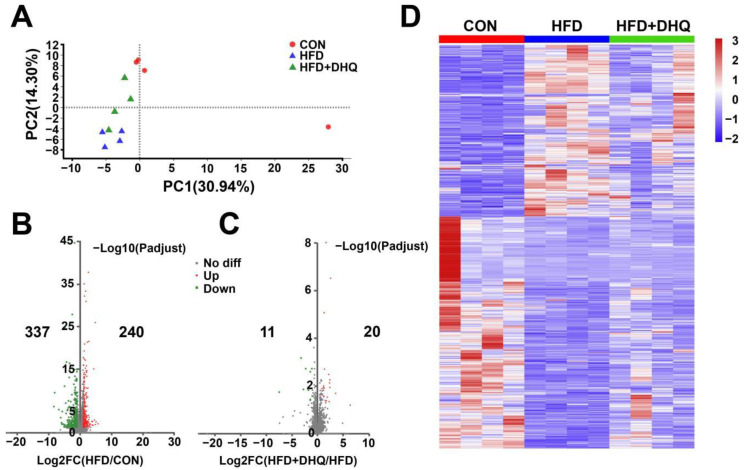
Effects of DHQ on hepatic genes expression in mice. (**A**) PCA score plots; (**B**) volcano plots of DEGs in CON vs. HFD; (**C**) volcano plots of DEGs in HFD vs. HFD+DHQ groups. Each dot represents a particular gene, and its corresponding horizontal and vertical values are gene expressive changes in the CON and HFD groups or HFD and HFD+DHQ groups, respectively. The red dots mean up-regulated genes expressed significantly, the green dots mean down-regulated genes expressed significantly, and the gray dots mean the genes expressed not significantly; and (**D**) heatmap of all DEGs altered by HFD and DHQ treatment.

**Figure 5 nutrients-14-05214-f005:**
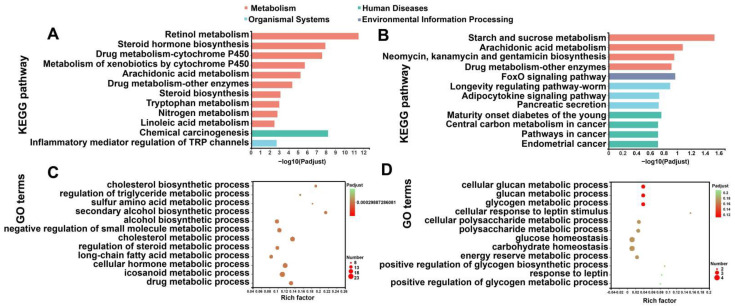
KEGG and GO enrichment analysis of altered gene sets in liver. (**A**) KEGG enrichment analysis of DEGs between CON and HFD groups; (**B**) KEGG enrichment analysis of DEGs between HFD and HFD+DHQ groups; (**C**) GO classification of DEGs between CON and HFD groups; and (**D**) GO classification of DEGs between HFD and HFD+DHQ groups. The vertical axis represents GO terms and the horizontal axis is the Rich factor. The size of each point indicates gene numbers in this GO Term.

**Figure 6 nutrients-14-05214-f006:**
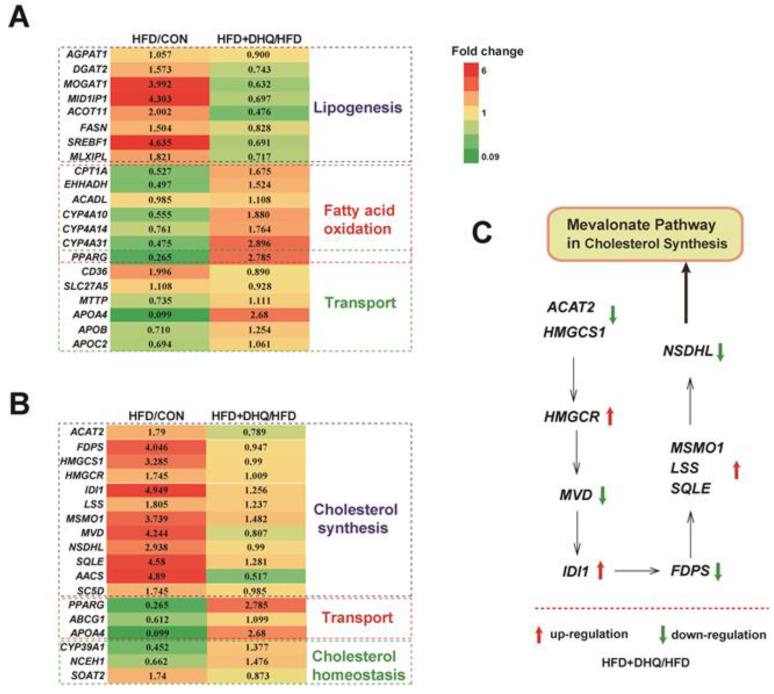
Effects of DHQ on fold change of lipid metabolic-related gene expression in liver in mice. (**A**) TG metabolism-related genes; (**B**) TC metabolism-related genes; (**C**) Mevalonate pathway in cholesterol synthesis. The red arrow represents up-regulation, and the green arrow represents down-regulation.

**Figure 7 nutrients-14-05214-f007:**
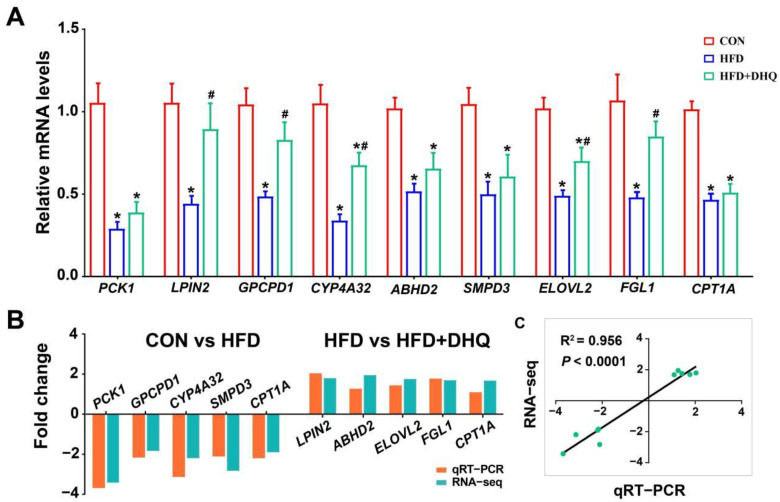
qRT-PCR validation of 9 DEGs selected randomly. (**A**) Relative mRNA levels of *PCK1*, *LPIN2*, *GPCPD1*, *CYP4A32*, *ABHD2*, *SMPD3*, *ELOVL2*, *FGL1*, and *CPT1A*; (**B**) fold change of up-regulation or down-regulation in RNA-seq and qRT-PCR assays; and (**C**) correlation analysis. Data were expressed as the mean ± SEM. * *p* < 0.05 vs. CON group, ^#^ *p* < 0.05 vs. HFD group.

**Figure 8 nutrients-14-05214-f008:**
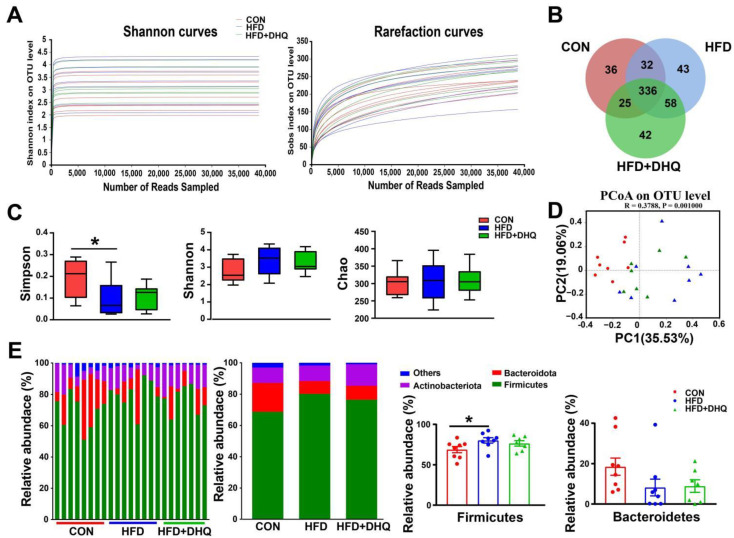
Effects of DHQ supplementation on fecal microbiota diversity and microbial composition at phylum level in mice. (**A**) Dilution curves; (**B**) Venn diagram; (**C**) α-diversity; (**D**) PCoA; and (**E**) microbial composition at phylum level. Data were expressed as the mean ± SEM. * *p* < 0.05 vs. CON group.

**Figure 9 nutrients-14-05214-f009:**
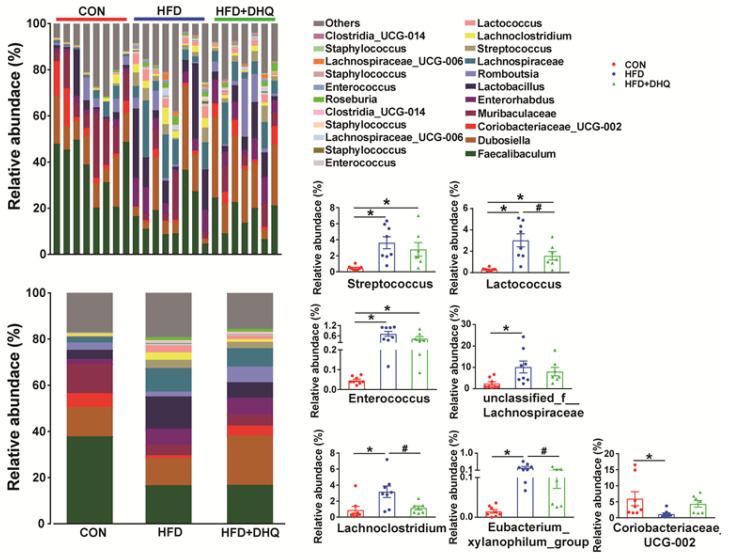
Effects of DHQ supplementation on fecal microbial composition at genus level in mice. Data were expressed as the mean ± SEM. * *p* < 0.05 vs. CON group, ^#^ *p* < 0.05 vs. HFD group.

**Figure 10 nutrients-14-05214-f010:**
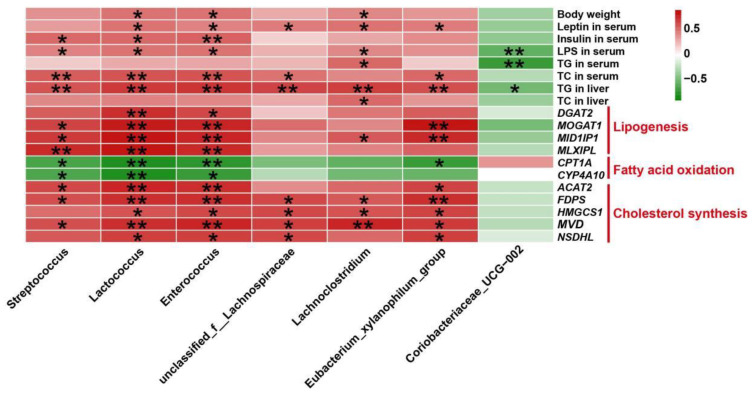
Pearson’s correlation analysis between fecal microbiota at the genus level and lipid metabolic indexes. * *p* < 0.05 and ** *p* < 0.01 represents significantly positive (red) or negative (green) correlation.

**Table 1 nutrients-14-05214-t001:** Primers applied for qRT-PCR detection.

Gene	Accession No.	Sequence (5′–3′)
*β-actin*	NM_007393.5	F: TGTCCACCTTCCAGCAGATGT R: GCTCAGTAACAGTCCGCCTAGAA
*PCK1*	NM_011044.3	F: CTACAACTTCGGCAAATACCT R: AACATCCACTCCAGCACCC
*LPIN2*	NM_001164885.1	F: CAGTGAAGACAGCCTCATAAGA R: GGAACAGGGTCTGCATCTAATA
*GPCPD1*	NM_001291050.1	F: GCTGTGATGCCCTGGGAAACTG R: TAGCGGTACTTCACTGACACTCCTC
*CYP4A32*	NM_001100181.1	F: TGGATTGGGTATGGTTTGC R: CACTGCCCTTGTGGCTGAA
*ABHD2*	NM_018811.6	F: CTGACCTCCCCACGAAT R: TTGCACACGATGTTACCAC
*SMPD3*	NM_021491.4	F: ATTGGTGGCGAGGAAGGAGGTC R: GCTGATTGTGGTTGGGTGTCTGG
*ELOVL2*	NM_001311121.1	F: ACCTTGTATAACCTCGCAATCA R: GAGATTCTGACACTGCAAGTTG
*FGL1*	NM_145594.2	F: CCAAGGAAACTGTGCTGAGGAAGAG R: TGCCCTGTAGGAACCACGGTAG
*CPT1A*	NM_013495.2	F: GATGTTCTTCGTCTGGCTTGA R: CTTATCGTGGTGGTGGGTGT

F, forward; R, reverse.

**Table 2 nutrients-14-05214-t002:** Quality check and comparative analysis results of RNA-seq results.

Sample	Raw Reads	Clean Reads	Q20 (%)	Q30 (%)	Total Mapped	Multiple Mapped	Uniquely Mapped
CON1	55,897,254	55,358,006	98.22	94.76	53,044,387 (95.82%)	4,460,339 (8.06%)	48,584,048 (87.76%)
CON2	56,224,080	55,658,230	98.2	94.7	53,369,649 (95.89%)	4,247,421 (7.63%)	49,122,228 (88.26%)
CON3	58,367,878	57,734,738	98.1	94.47	55,392,457 (95.94%)	4,322,053 (7.49%)	51,070,404 (88.46%)
CON4	59,835,300	59,222,546	98.24	94.82	56,606,223 (95.58%)	4,836,943 (8.17%)	51,769,280 (87.41%)
HFD1	54,072,972	53,494,468	98.14	94.58	50,917,010 (95.18%)	4,501,150 (8.41%)	46,415,860 (86.77%)
HFD2	54,066,716	53,486,188	98.12	94.51	51,426,931 (96.15%)	3,914,020 (7.32%)	47,512,911 (88.83%)
HFD3	54,408,164	53,793,496	98.15	94.6	51,614,914 (95.95%)	3,920,299 (7.29%)	47,694,615 (88.66%)
HFD4	50,821,272	50,323,172	98.22	94.76	47,772,983 (94.93%)	4,033,410 (8.02%)	43,739,573 (86.92%)
HFD+DHQ1	56,828,852	56,138,872	98.09	94.43	53,095,738 (94.58%)	4,810,153 (8.57%)	48,285,585 (86.01%)
HFD+DHQ2	55,367,108	54,854,566	98.24	94.81	52,096,234 (94.97%)	4,635,712 (8.45%)	47,460,522 (86.52%)
HFD+DHQ3	53,409,378	52,932,284	98.34	95.08	50,464,426 (95.34%)	4,443,960 (8.4%)	46,020,466 (86.94%)
HFD+DHQ4	56,021,494	55,494,786	98.29	94.96	53,368,642 (96.17%)	4,126,675 (7.44%)	49,241,967 (88.73%)

## Data Availability

All raw sequences from hepatic transcriptomics and 16S rRNA gene-based analysis were uploaded to NCBI Sequence Read Archive (SRA) database (Accession Number: PRJNA885265 and PRJNA882752).

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
