# Peer review of "Dihydroquercetin Supplementation Improved Hepatic Lipid Dysmetabolism Mediated by Gut Microbiota in High-Fat Diet (HFD)-Fed Mice"

_nutrients, 2022, doi:10.3390/nu14245214_

Round 1
Reviewer 1 Report
In this manuscript, the authors have investigated whether dihydroquercetin (DHQ) supplementation improved lipid dysmetabolism in the body, especially in the liver. While I appreciate the effort of the work presented and the significance of comparing the relationship between hepatic metabolism and altered gut microbiota, I think the authors needs to improve some data of the paper and provide comprehensive discussion content. I would like the authors to consider the following correction.
1. In Figure 1, It is better to separate the figures by A-I and J-K. A-I is the result of some weight measurement and its index, while J-K is the result of analysis based on experiment. Since the quality of these data is different, they should be separated as figures.
2. Differences in the results of HE staining images in Figure 1F are difficult to distinguish between mice groups. Images at 10x magnification would be unnecessary, and images at 20x and 40x magnification would be more suitable.
3. In Figure 2, the evaluation of serum lipids is sufficient, but the data on liver function are insufficient. Whether DHQ really improves liver function is critical in this study, and indexes such as AST, ALT, and γ-GT in serum should be measured.
4. In Figure 7 and 8, considerable differences in gut microbiota are observed even among mice in the Con group. Therefore, I feel that the analysis of the intestinal microflora is highly dependent on individual differences and that the differences between the three groups in this study may be meaningless. I think that the chronological changes of the intestinal microflora in the same mouse are more important. Do the authors have any opinions on these points?
5. Since the metabolism of DHQ in vivo has not been shown in this study, the relevance of each result in the liver and gut microbiota is not well discussed. In figures 4-6, DHQ improves lipid metabolism parameters in the liver, so does this mean that DHQ reaches liver and works in there?Similarly, in figures 7-8, do the components of DHQ really reach the gut and improve the intestinal microflora? If DHQ is already absorbed or disassembled in the stomach, I think it has not enough influence to regulate the gut microbiota. Have the authors confirmed the biodistribution of DHQ in mice body (stomach, liver, small intestine, colon, and serum, etc…)? How long does DHQ remain in vivo to be effective? Additional experiments are needed to determine DHQ concentrations in various organs or to discuss the disposition of DHQ in the mice body with reference to previous papers.
6. As a result of DHQ administration, which comes first, inhibiting the biosynthesis of lipid or improvement of gut microbiota? This opinion significantly changes the interpretation of the results. In the discussion of this manuscript, the cause and result relationship between these two events is very difficult to understand. According to the final paragraph of the discussion, the authors seem to insist that the improved intestinal microbiota results in suppression of lipid levels; however they do not clearly state their relevance in conclusion section.
7. In the discussion, the authors emphasize only the advantages of DHQ intake with referring to a lot of publications, but they do not mention the disadvantages. Are there any side effects or addiction symptoms from an overdose of DHQ? What is the recommended DHQ intake for health benefits? The authors should express their opinions on these matters with referring to previous papers.
The animal experiment design itself is good, but I think that the results are not well discussed and conclusions are not meaningful in this manuscript. While the findings presented in this study should be of interest to the audience of this journal, it is my opinion that a rather substantial revision, based on the comments given above, is needed to make this manuscript suitable for publication.
Reviewer 2 Report
The authors have used 16S rRNA sequencing methods to maps the modulation of hepatic cholesterol biosynthesis gene and gut microbiota composition by dihydroquercetin (DHQ) supplementation. The further positive correlation between gut bacteria and lipid and cholesterol biosynthesis gene has been displayed. Their detailed work showed how DHQ supplementation improved hepatic lipid dysmetabolism mediated by gut microbiota in high-fat diet fed mice.
Line 201 mentioned about energy intake. No description about how to evaluate energy intake in the methodology section.
Please provide the raw data for Figure 1 B-D, I-K (eight mice’s various parameters) in the support information.
Round 2
Reviewer 1 Report
The manuscript has been improved and is in a nice condition now. I think that the authors have carefully researched previous publications, and have conducted their study with sufficient knowledge. Most importantly, this study represents a massive effort and it deserves to be published somewhere. I consider them to have done a good work.